# Effect size estimates from umbrella designs: Handling patients with a positive test result for multiple biomarkers using random or pragmatic subtrial allocation

**Miriam Kesselmeier** [1,2]ℰ *, **Norbert Benda**[3,4]ℰ, **André Scherag**[2]ℰ

**1** Research Group Clinical Epidemiology, Center for Sepsis Control and Care (CSCC), Jena University Hospital, Jena, Germany, **2** Institute of Medical Statistics, Computer and Data Sciences, Jena University Hospital, Jena, Germany, **3** Research Department, Federal Institute for Drugs and Medical Devices, Bonn, Germany, **4** Department of Medical Statistics, University Medical Center Göttingen, Göttingen, Germany

ℰ These authors contributed equally to this work.
* miriam.kesselmeier@med.uni-jena.de

**Data Availability Statement:** The R source code for the analytical calculations as well as for the simulation study is available as supplemental

## Abstract

Umbrella trials have been suggested to increase trial conduct efficiency when investigating different biomarker-driven experimental therapies. An overarching platform is used for patient screening and subsequent subtrial allocation according to patients' biomarker status. Two subtrial allocation schemes for patients with a positive test result for multiple biomarkers are (i) the pragmatic allocation to the eligible subtrial with the currently fewest included patients and (ii) the random allocation to one of the eligible subtrials. Obviously, the subtrials compete for such patients which are consequently underrepresented in the subtrials. To address questions of the impact of an umbrella design in general as well as with respect to subtrial allocation and analysis method, we investigate an umbrella trial with two parallel group subtrials and discuss generalisations. First, we analytically quantify the impact of the umbrella design with random allocation on the number of patients needed to be screened, the biomarker status distribution and treatment effect estimates compared to the corresponding gold standard of an independent parallel group design. Using simulations and real data, we subsequently compare both allocation schemes and investigate weighted linear regression modelling as possible analysis method for the umbrella design. Our results show that umbrella designs are more efficient than the gold standard. However, depending on the biomarker status distribution in the disease population, an umbrella design can introduce differences in estimated treatment effects in the presence of an interaction between treatment and biomarker status. In principle, weighted linear regression together with the random allocation scheme can address this difference though it is difficult to assess if such an approach is applicable in practice. In any case, caution is required when using treatment effect estimates derived from umbrella designs for e.g. future trial planning or meta-analyses.

material. The authors were given access to the patient data of the real data application. The SepNet Study Group owns this data (or to be more correct has the right to process the data along the lines of the written informed consent). Due to restricted access, the data from the real data application cannot be provided by the authors – we refer to the SepNet Study Group (https://www.sepsis-stiftung.eu/sepnet/; study coordination: office@sepsisstiftung.de) for data access questions. The authors had no special access privileges others would not have to the data.

**Funding:** This work (MK, AS) was supported by the Integrated Research and Treatment Center, Center for Sepsis Control and Care (CSCC), at the Jena University Hospital funded by the German Ministry of Education and Research (BMBF No. 01EO1502). AS also received funding by BMBF No. 01ZZ1803C. The funders had no role in the study design, data collection and analysis, decision to publish, or preparation of the manuscript. The URL of the funder website is https://www.bmbf.de/en/index.html.

**Competing interests:** The authors have declared that no competing interests exist.

## Introduction

The gold standard trial design to investigate experimental treatments is the conduct of independent parallel group trials, i.e. one trial for each experimental treatment (upper panel of Fig 1). Ideally, such independent trials do not influence each other during trial conduct, i.e. patients are always eligible for only one of these independent trials. To increase trial conduct efficiency, a platform trial uses a platform to screen patients for several subtrials. If these subtrials investigate different biomarker-driven experimental treatments within one disease type, such a platform trial is called "umbrella trial" (lower panel of Fig 1; e.g. [1, 2]). The combined individual test results for the biomarkers under investigation (biomarker status) are used to allocate the patient to a subtrial. Conceptually, umbrella trials have been extended to the option of stopping or adding subtrials related to targeted treatments—especially in oncology (e.g. [3–6]).

A challenge of umbrella designs that investigate single targeted treatments arises if patients have a positive test result for multiple biomarkers under investigation. In this case, the patient is eligible for more than one subtrial. However, this patient can (usually) only be allocated to one of the subtrials. Hence, such patients will be underrepresented in some (or even all) umbrella subtrials. Further hampering the interpretation, the scheme for allocating such patients to subtrials may also vary. Despite pragmatism and plausibility of the umbrella design idea, we realised that little is known about statistical properties of umbrella trials and possible allocation schemes.

This study has three objectives. First, we analytically quantify the impact of conducting an umbrella trial compared to the corresponding gold standard of independent parallel group trials (Section "Impact of the umbrella design with the random allocation scheme"; details about

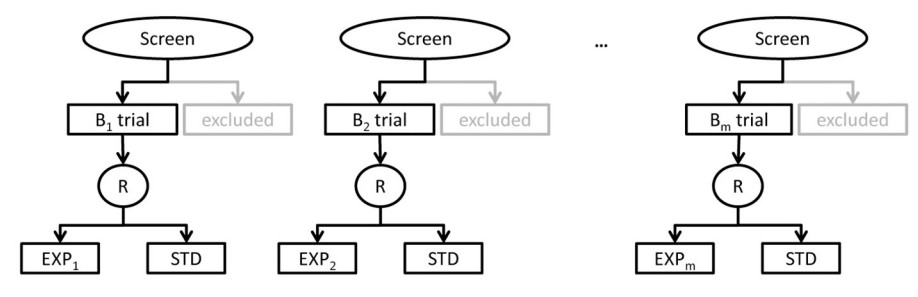

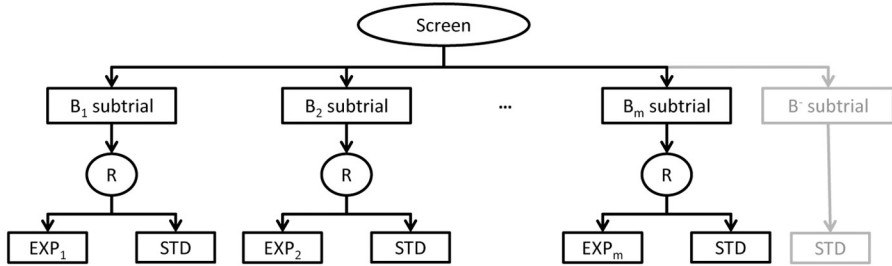

**Fig 1. Trial designs.** An example for independent parallel group trials (gold standard trial design; upper panel) and the corresponding umbrella parallel group subtrials (lower panel). All patients in the $B_i$ trial and in the $B_i$ subtrial, respectively, exhibit a positive test result for biomarker $B_i$ ($i = 1, \ldots, m$). Patients in the $B^-$ subtrial exhibit negative test results for all $B_i$. Screen: screening platform, R: randomisation, $EXP_i$: experimental treatment related to $B_i$, STD: standard treatment.

the investigated study design scenarios are provided in the Section "Investigated study design scenarios"). To quantify design differences, we compare the number of patients needed to be screened, the biomarker status distribution and the treatment effect estimates. Secondly, we contrast two subtrial allocation schemes in a simulation study and in real data with respect to the number of patients needed to be screened and biomarker status distribution (Section "Impact of the pragmatic subtrial allocation scheme and the analysis method"). Thirdly, we propose possible analysis solutions to obtain similar treatment effect estimates for the gold standard and the umbrella design based on simulations and on real data (also Section "Impact of the pragmatic subtrial allocation scheme and the analysis method"). Finally, we discuss our results, derive practical recommendations and generalisations beyond the two-biomarker setting and their translation to other master protocols (Section "Discusion"). R source code for the analytical derivations and for the simulation study is available as S1 File.

## Investigated study design scenarios

As in Fig 1, we assume one standard treatment—which could also be placebo—and two experimental targeted treatments linked to two different, binary biomarkers $B_i$ ($i$ = 1, 2). Let $\pi_i$ be the prevalence of a positive test result for $B_i$. We denote the patient's combined test results of all biomarkers as the patient's biomarker status. Let $B_i^+$ be an indicator for a positive test result for $B_i$. Then, the tuple $(B_1^+, B_2^+)$ denotes the biomarker (positive) status of a patient and $P[(B_1^+, B_2^+)]$ the probability of having a specific biomarker status as a screened but not yet included patient. This probability corresponds to the expected biomarker status distribution in the (disease) population. This distribution is given by

$$
\begin{aligned}
P[(1,1)] &= \pi_1 \pi_2 + \phi \sqrt{\pi_1 (1 - \pi_1) \pi_2 (1 - \pi_2)} \\
P[(1,0)] &= \pi_1 - P[(1,1)] \\
P[(0,1)] &= \pi_2 - P[(1,1)] \\
P[(0,0)] &= 1 - \pi_1 - \pi_2 + P[(1,1)]
\end{aligned}
\tag{1}
$$

with the $\phi$ coefficient as correlation (dependency) measure between the two biomarkers ($\phi \in [-1, 1]$). For a randomly selected patient, $\phi$ corresponds to the correlation between the binary random variables $B_1^+$ and $B_2^+$, i.e.

$$
\phi = \frac{E[B_1^+] E[B_2^+] - E[B_1^+ B_2^+]}{\sqrt{Var[B_1^+] Var[B_2^+]}}
\tag{2}
$$

with expectation $E[\cdot]$ and variance $Var[\cdot]$.

To evaluate the efficacy of the experimental treatments, each experimental treatment is compared to the standard treatment with a 1:1 treatment arm allocation scheme for a normally distributed outcome. We consider an independent (randomised controlled) trial design as well as corresponding umbrella trial designs. Hereinafter, we will use the term "(sub-) trial" for combined consideration of both the independent trial and the corresponding umbrella subtrial. The patient's (sub-) trial allocation is provided in the Subsections "Independent trial design" and "Umbrella trial design". The (sub-) trials are analysed separately and comparisons between the (sub-) trials (e.g. comparisons of experimental treatments) are not intended. Other analyses strategies—in particular for umbrella trials—are discussed in the Section "Discussion". To analyse each (sub-) trial,

we consider un-weighted and weighted mean group differences. While the former represents a standard measure, weighting is a suggestion to generate similar treatment effect estimates for the independent trial and the corresponding umbrella subtrial by mimicking the biomarker status distribution of the independent trial. Details on the weighted regression approach and on the calculation of the weights in a practical application (with two or more subtrials) are provided in S2 Note.

### Independent trial design

As gold standard, we consider two independent parallel group trials which we denote "trial 1" and "trial 2". We assume that the trials do not compete for eligible patients. This means that the trials run independently, a trial cannot be influenced by the respective other trial and, hence, the patient's trial allocation is biunique. Patients are eligible for and allocated to trial $i$ if they exhibit a positive test result for biomarker $B_i$ and discarded otherwise. Trial $i$ recruits until its planned total sample size $N_i$ is reached.

### Umbrella trial design

We compare the gold standard of the Subsection "Independent trial design" to an umbrella trial with two parallel group subtrials which we denote "subtrial 1" and "subtrial 2". Let both subtrials start at the same point in time. Subtrial $i$ includes patients until its planned subtrial sample size of $N_i$ is reached. The umbrella trial closes if recruitment for both subtrials is completed.

In an umbrella trial, patients with a positive test result for biomarker $B_i$ are eligible for subtrial $i$. Patients with a positive test result for $B_i$ only are allocated to subtrial $i$ if this subtrial is still recruiting and discarded otherwise. Unlike the independent trial design, the two subtrials compete for patients with a positive test result for both biomarkers due to the joint screening platform. We investigate two subtrial allocation schemes for such patients that have been suggested: random and pragmatic allocation [2, 7]. Under the random allocation scheme, allocation of a patient with a double positive test result to one of the two subtrials happens with equal probability. Under the pragmatic allocation scheme, such a patient is allocated to the subtrial that has included fewer patients at the time of allocation. In case of an already completed recruitment for one subtrial, patients with a double positive test result are allocated to the still recruiting subtrial for both allocation schemes. In the Section "Discussion", we discuss generalisations to more than two biomarkers/experimental treatments.

### Impact of the umbrella design with the random allocation scheme

In this section, we derive analytical solutions for the number of patients needed to be screened, the biomarker status distribution and the expected treatment effect for the independent trials and the umbrella subtrials with the random allocation scheme. These formulae are subsequently illustrated.

### Number of patients needed to be screened

The expected number of patients $E[N_{screen}]$ that need to be screened is a measure that can be used to compare trial designs. The number of patients needed to be screened in an independent trial follows a negative binomial distribution, and, hence, for the independent trial design,

the total number across both trials is

$$E[N_{screen}] = \frac{N_1}{\pi_1} + \frac{N_2}{\pi_2} \tag{3}$$

for given trial size $N_i$ and prevalence $\pi_i$ of a positive test result for biomarker $B_i$ [8]. This holds because the two trials are independent.

For the umbrella design with the random allocation scheme, the subtrials compete for patients with a positive test result for both biomarkers. Consequently, the prevalence of a positive test result for biomarker $B_k$ influences the expected number of patients that must be screened for subtrial $i$ ($i \neq k$) and vice versa. Let $r \in [0, 1]$ denote the proportion of patients with a positive test result for both biomarkers that are allocated to subtrial 1 when both subtrials are recruiting (here: $r = 0.5$). Let $q_0$ denote the proportion of patients in the (disease) population that are eligible for the trial, i.e. exhibit a positive test result for at least one biomarker. It is

$$q_0 = 1 - P[(0,0)] \tag{4}$$

Let $p_1^{(1)}$ denote the probability that a patient who is eligible for the umbrella trial is included in subtrial 1 when both subtrials are recruiting. It is

$$p_1^{(1)} = \frac{\pi_1 - (1-r)\,P[(1,1)]}{q_0} \tag{5}$$

Let $E_i^{(1)}$ denote the expected number of patients that are included in the trial during the time segment when patients are recruited for both subtrials in case that subtrial $i$ closes earlier than subtrial $k$ ($i, k = 1, 2, i \neq k$). Then,

$$
\begin{aligned}
E_1^{(1)} &= \left(1 + \frac{1 - p_1^{(1)}}{p_1^{(1)}} \frac{I_{p_1^{(1)}}(N_1 + 1, N_2 - 1)}{I_{p_1^{(1)}}(N_1, N_2)}\right) N_1 \\
E_2^{(1)} &= \left(1 + \frac{p_1^{(1)}}{1 - p_1^{(1)}} \frac{I_{1-p_1^{(1)}}(N_2 + 1, N_1 - 1)}{I_{1-p_1^{(1)}}(N_2, N_1)}\right) N_2
\end{aligned}
\tag{6}
$$

with the regularized incomplete beta function $I_p(\cdot, \cdot)$ for $p \in \{p_1^{(1)}, 1 - p_1^{(1)}\}$. Analogously, let $E_i^{(2)}$ denote the expected number of patients with a positive test result for at least one biomarker during the time segment where subtrial $i$ is already closed. Then,

$$
\begin{aligned}
E_1^{(2)} &= \frac{q_0}{\pi_2}\left(N_2 - \frac{1 - p_1^{(1)}}{p_1^{(1)}} \frac{I_{p_1^{(1)}}(N_1 + 1, N_2 - 1)}{I_{p_1^{(1)}}(N_1, N_2)} N_1\right) \\
E_2^{(2)} &= \frac{q_0}{\pi_1}\left(N_1 - \frac{p_1^{(1)}}{1 - p_1^{(1)}} \frac{I_{1-p_1^{(1)}}(N_2 + 1, N_1 - 1)}{I_{1-p_1^{(1)}}(N_2, N_1)} N_2\right)
\end{aligned}
\tag{7}
$$

Let $X$ be the number of patients included in subtrial 2 at the moment of subtrial 1's closing and $Y$ the number of patients included in subtrial 1 at the moment of subtrial 2's closing. Then, $P[X < N_2]$ and $P[Y < N_1] = 1 - P[X < N_2]$ denote the probability that subtrial 1 closes

earlier than subtrial 2 and vice versa, respectively. It holds that

$$
\begin{aligned}
P[X < N_2] &= I_{p_1^{(1)}}(N_1, N_2) \\
P[Y < N_1] &= I_{1-p_1^{(1)}}(N_2, N_1)
\end{aligned}
\tag{8}
$$

Then, the expected total number of screened patients for the umbrella trial is given by

$$
E[N_{\text{screen}}] = \left\{ \left( E_1^{(1)} + E_1^{(2)} \right) P[X < N_2] + \left( E_2^{(1)} + E_2^{(2)} \right) P[Y < N_1] \right\} \frac{1}{q_0}
\tag{9}
$$

For details on the derivation we refer to S1 Note.

A comparison of Eq (9) with Eq (3) indicates that fewer patients are discarded by an umbrella trial compared to the independent trial design. The gain in efficiency of an umbrella trial depends on the prevalence of a positive test for the biomarkers, i.e. an increasing similarity in the prevalence of the biomarkers induces an increased efficiency gain with respect to the number of discarded patients because the time span where patients are recruited for both subtrials is larger in this case.

## Biomarker status distribution

The expected biomarker status distribution in the (disease) population as provided in Eq (1) translates into the distributions in the (sub-) trials. The (sub-) trial-specific distribution is related to the conditional probability for a specific biomarker status given the considered (sub-) trial. For the independent trials, these conditional probabilities are

$$
\begin{aligned}
P[(1,0)|\text{trial } 1] &= 1 - \frac{P[(1,1)]}{\pi_1} \\
P[(1,1)|\text{trial } 1] &= \frac{P[(1,1)]}{\pi_1} \\
P[(1,1)|\text{trial } 2] &= \frac{P[(1,1)]}{\pi_2} \\
P[(0,1)|\text{trial } 2] &= 1 - \frac{P[(1,1)]}{\pi_2}
\end{aligned}
\tag{10}
$$

For the umbrella design with the random allocation scheme, let $p_{0,i}^{(1)}$ denote the probability that a patient is allocated to subtrial $i$ in case both subtrials are recruiting, i.e.

$$
\begin{aligned}
p_{0,1}^{(1)} &= \pi_1 - (1 - r)\, P[(1,1)] \\
p_{0,2}^{(1)} &= \pi_2 - r\, P[(1,1)]
\end{aligned}
\tag{11}
$$

Then, the conditional probabilities for the umbrella subtrials are

$$
P[(1,0)|\text{subtrial } 1] = \frac{P[(1,0)]}{N_1}\left\{ \frac{N_1}{p_{0,1}^{(1)}} P[X < N_2] \right.
$$

$$
\left. + \left( \frac{E_2^{(1)} - N_2}{p_{0,1}^{(1)}} + \frac{E_2^{(2)}}{q_0} \right) P[Y < N_1] \right\}
$$

$$
P[(1,1)|\text{subtrial } 1] = \frac{P[(1,1)]}{N_1}\left\{ \frac{r\, N_1}{p_{0,1}^{(1)}} P[X < N_2] \right.
$$

$$
\left. + \left( \frac{r\,(E_2^{(1)} - N_2)}{p_{0,1}^{(1)}} + \frac{E_2^{(2)}}{q_0} \right) P[Y < N_1] \right\}
$$

$$
P[(1,1)|\text{subtrial } 2] = \frac{P[(1,1)]}{N_2}\left\{ \frac{(1-r)\, N_2}{p_{0,2}^{(1)}} P[Y < N_1] \right. \tag{12}
$$

$$
\left. + \left( \frac{(1-r)\,(E_1^{(1)} - N_1)}{p_{0,2}^{(1)}} + \frac{E_1^{(2)}}{q_0} \right) P[X < N_2] \right\}
$$

$$
P[(0,1)|\text{subtrial } 2] = \frac{P[(0,1)]}{N_2}\left\{ \frac{N_2}{p_{0,2}^{(1)}} P[Y < N_1] \right.
$$

$$
\left. + \left( \frac{E_1^{(1)} - N_1}{p_{0,2}^{(1)}} + \frac{E_1^{(2)}}{q_0} \right) P[X < N_2] \right\}
$$

For details on the derivations we refer to S1 Note.

For given biomarker status $(B_1^+, B_2^+)$ and (sub-) trial $i$, the difference in the biomarker status distribution between the umbrella subtrial $i$ and the corresponding independent trial $i$ is given by

$$
\Delta P_{(B_1^+, B_2^+),i} := P[(B_1^+, B_2^+) \mid \text{subtrial } i] - P[(B_1^+, B_2^+) \mid \text{trial } i] \tag{13}
$$

The results provided in Eqs (10) and (12) indicate that the proportion of patients with a double positive test result is smaller in the umbrella subtrials relative to the corresponding independent trials. The prevalence $\pi_k$ ($i \neq k$) drives this proportion in (sub-) trial $i$. It also becomes clear that the distributions depend on the recruitment of the subtrials, i.e. the duration in which only one subtrial is recruiting. Consequently, the similarity of the distributions in subtrial $i$ and trial $i$ depends on the difference in recruitment time between the subtrials. The larger this difference, the more similar are the distributions of the longer recruiting subtrial and the corresponding trial and vice versa for the other (sub-) trial.

## Estimated treatment effect

Based on the differences in the biomarker status distribution between corresponding independent trial and umbrella subtrial, we may also expect an impact on the estimated treatment effects if treatment effects differ between the different subpopulations defined by the four possible biomarker status, i.e. in the presence of an interaction between biomarker status and treatment. More precisely, if the treatment effect in patients with a positive test result for $B_1$

depends on the status of the other biomarker $B_2$ (or vice versa), the effect estimation in subtrial 1 (and subtrial 2, respectively) of the umbrella design will differ from results derived in the corresponding independent trial due to the underrepresentation of patients with a positive test result for both biomarkers.

Let $\delta_i$ denote the expected treatment effect against a standard treatment group in the independent trial $i$. Furthermore, let $\delta_{(B_1^+, B_2^+),i}$ denote the corresponding biomarker status-dependent expected treatment effect in trial $i$. Then, the difference between the expected treatment effect in the umbrella subtrial $i$ and in the corresponding independent trial $i$—denoted by $\Delta X_i$—is given by

$$\Delta X_1 := \Delta P_{(1,0),1}\, \delta_{(1,0),1} + \Delta P_{(1,1),1}\, \delta_{(1,1),1} \quad \text{for subtrial 1 against trial 1}$$

$$\Delta X_2 := \Delta P_{(0,1),2}\, \delta_{(0,1),2} + \Delta P_{(1,1),2}\, \delta_{(1,1),2} \quad \text{for subtrial 2 against trial 2}$$

(14)

with $\Delta P_{(B_1^+, B_2^+),i}$ from Eq (13).

Accordingly, differences in estimated treatment effects between independent trials and umbrella subtrials depend on both differences in the biomarker status distributions and on the magnitude by which the treatment effects depends on the biomarker status. A difference in treatment effect estimates between corresponding independent trials and umbrella subtrials only arises if a patient's response to the $B_i$-related treatment depends on the patient's $B_k$ status ($i \neq k$). Both larger and smaller treatment effect estimates may result for umbrella subtrials relative to the corresponding independent trial. Obviously, such differences also have an impact on the statistical (error) probabilities of test statistics because it changes the difference between the estimated value and the expected value under the null hypothesis. Thus, both type I and type II error probabilities can be affected. In other words, type I error and hence the validity of a confirmatory conclusion from subtrial $i$ will be affected in the presence of an interaction between treatment and biomarker status in patients with a positive test result for $B_i$.

## Illustration of the analytical derivations

To illustrate practically the analytical derivations, we choose prevalence constellations from studies on breast cancer [9, 10] and pharmacogenomics, e.g. [11]. Let be

$$\pi_1 \in \begin{cases} \{0.012, 0.024, 0.120, 0.240\} & \text{for } \pi_2 = 0.250 \\ \{0.012, 0.024\} & \text{for } \pi_2 = 0.025 \end{cases}$$

We denote a prevalence of a positive test result for biomarker $B_i$ "low" if $\pi_i \in \{0.012, 0.024, 0.025\}$ and "high" otherwise. Furthermore, we assume independence of the biomarkers ($\phi = 0$), equal sized (sub-) trials ($N_1 = N_2 =: N$), equal sized treatment arms (each of size $0.5N$) and equal treatment effects in the trials ($\delta_1 = \delta_2 =: \delta$). In practice, these assumptions will probably not hold, especially for small sample sizes, but this is not relevant for this illustration. Then, we assume for the treatment effect $\delta$ that (i) a larger individual outcome is equivalent to a better patient's response to the treatment, (ii) patients with a single positive test result respond better (in terms of assumption (i)) to treatment than those with a double positive test result and (iii) it can be detected by Student's t-test with a comparison-wise power of 0.8 at a two-sided significance level of 0.05 for a common standard deviation of 1 in an independent trial for given trial size N. We refer to the Section "Discussion" for an in-depth discussion of these

**Table 1. Assumed biomarker status- and (sub-) trial-specific individual outcomes.**

| Biomarker status $(B_1^+, B_2^+)$ | (Sub-) trial 1 | | (Sub-) trial 2 | |
|---|---|---|---|---|
| | experimental treatment | standard treatment | experimental treatment | standard treatment |
| (1, 0) | $\mu_{(1,0),1,EXP}$ | 0.2 | – | – |
| (1, 1) | $-\frac{\delta}{2}$ | 0.0 | $\frac{\delta}{2}$ | 0.0 |
| (0, 1) | – | – | $\mu_{(0,1),2,EXP}$ | 0.2 |

$B_i$ ($i$ = 1, 2) denotes the biomarker $i$, $B_i^+$ is an indicator for a positive test result for biomarker $B_i$ and $(B_1^+, B_2^+)$ denotes the biomarker status. Let $\delta$ denote the given expected treatment effect in the independent trials. $\mu_{(1,0),1,EXP}$ and $\mu_{(0,1),2,EXP}$ denote the expected outcome of patients with a single positive test result receiving the experimental treatment in (sub-) trial 1 and 2, respectively, and are provided in Eqs (15) and (16). All other values are arbitrarily fixed.

assumptions. Thus, the treatment effect $\delta$ is

$$\delta = \begin{cases} 0.57 & \text{for } N = 100 \\ 0.36 & \text{for } N = 250 \end{cases}$$

detectable with the given N. If we now arbitrarily fix parts of the expected treatment effect as provided in Table 1 such that $\delta$ results for each independent trial, we can solve

$$\begin{aligned} \delta \quad &= \mu_{1,EXP} - \mu_{1,STD} \\[2mm] &= \left( \mu_{(1,0),1,EXP} \, P[(1,0)|\text{trial } 1] + \left( -\frac{\delta}{2} \right) P[(1,1)|\text{trial } 1] \right) \\[2mm] &\quad - (0.2 \, P[(1,0)|\text{trial } 1] + 0 \, P[(1,1)|\text{trial } 1]) \end{aligned} \quad (15)$$

for $\mu_{(1,0),1,EXP}$ in the independent trial 1 and

$$\begin{aligned} \delta \quad &= \mu_{2,EXP} - \mu_{2,STD} \\[2mm] &= \left( \mu_{(0,1),2,EXP} \, P[(0,1)|\text{trial } 2] + \frac{\delta}{2} \, P[(1,1)|\text{trial } 2] \right) \\[2mm] &\quad - (0.2 \, P[(0,1)|\text{trial } 2] + 0 \, P[(1,1)|\text{trial } 2]) \end{aligned} \quad (16)$$

for $\mu_{(0,1),2,EXP}$ in the independent trial 2 with $\mu_{i,EXP}$ and $\mu_{i,STD}$ denoting the expected outcome in the independent trial $i$ for patients receiving the experimental and standard treatment, respectively. Here, $\mu_{(B_1^+, B_2^+),i,EXP}$ denotes the expected outcome in the independent trial $i$ for patients with biomarker status $(B_1^+, B_2^+)$ receiving the experimental treatment. Note that the impact of the biomarker status is larger in (sub-) trial 1 induced by the reverse effect on patients with a positive test result for both biomarkers. This assumption may be unrealistic in practice but is useful for this illustration.

The first aspect of the analytical derivations is the number of screened patients. A measure of efficiency of a trial design is the ratio of discarded (screened but not included) to included patients (see Fig 2). Of course, this ratio varies with the prevalence of positive test results for the biomarkers. If the prevalence of positive test results is low for both biomarkers or for both high, the application of an umbrella design reduces the ratio by one third to one half compared to the independent trial design. Otherwise, it reduces the ratio by 4 to 8%.

The second aspect of the analytical investigation is the change in the biomarker status distribution introduced by the umbrella design (see Fig 3). If the prevalence of a positive test result is low for biomarker $B_1$ and high for $B_2$, the frequency of patients with a positive test

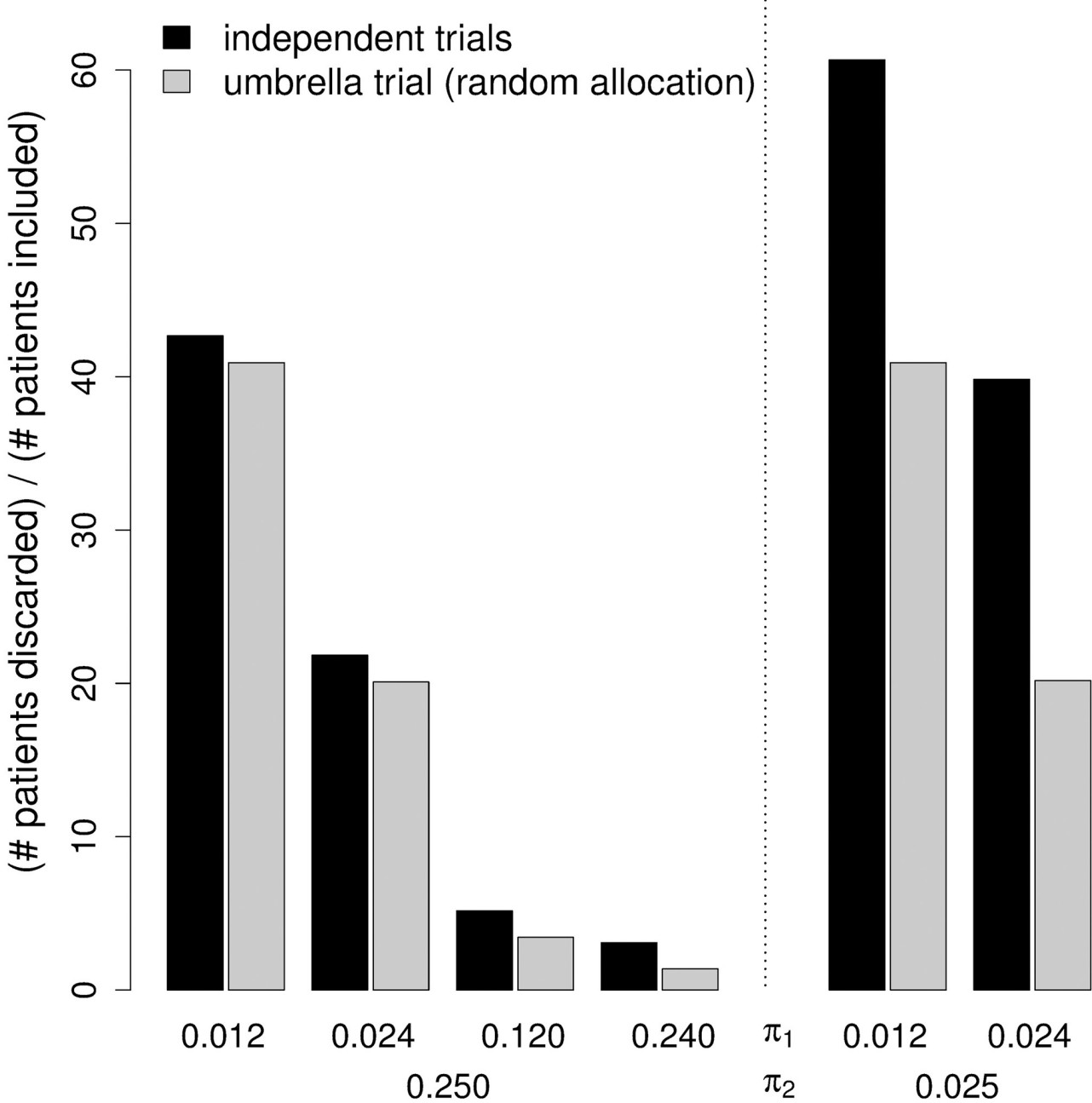

**Fig 2. Ratio of discarded to included patients in the independent trial design and in the umbrella design with the random allocation scheme.** $\pi_i$ ($i$ = 1, 2) denotes the prevalence of a positive test result for biomarker $B_i$. A "discarded patient" is a patient that was screened but not included in a (sub-) trial. The ratio is derived as $E[N_{screen}]-2N$ divided by $2N$ with $E[N_{screen}]$ from Eqs (3) and (9), respectively. #: number of.

result for both biomarkers in subtrial 1 decreases by 1 to 4% compared to trial 1. Otherwise, the decrease in subtrial 1 is between 19 and 48%. In subtrial 2, we observe a reduction by 40 to 50% compared to trial 2. Obviously, this is directly related to the probability that a specific subtrial closes earlier than the other subtrial (see Fig 4). The question about which subtrial is expected to close first can be answered biunique except for the case of two very similar prevalent biomarkers.

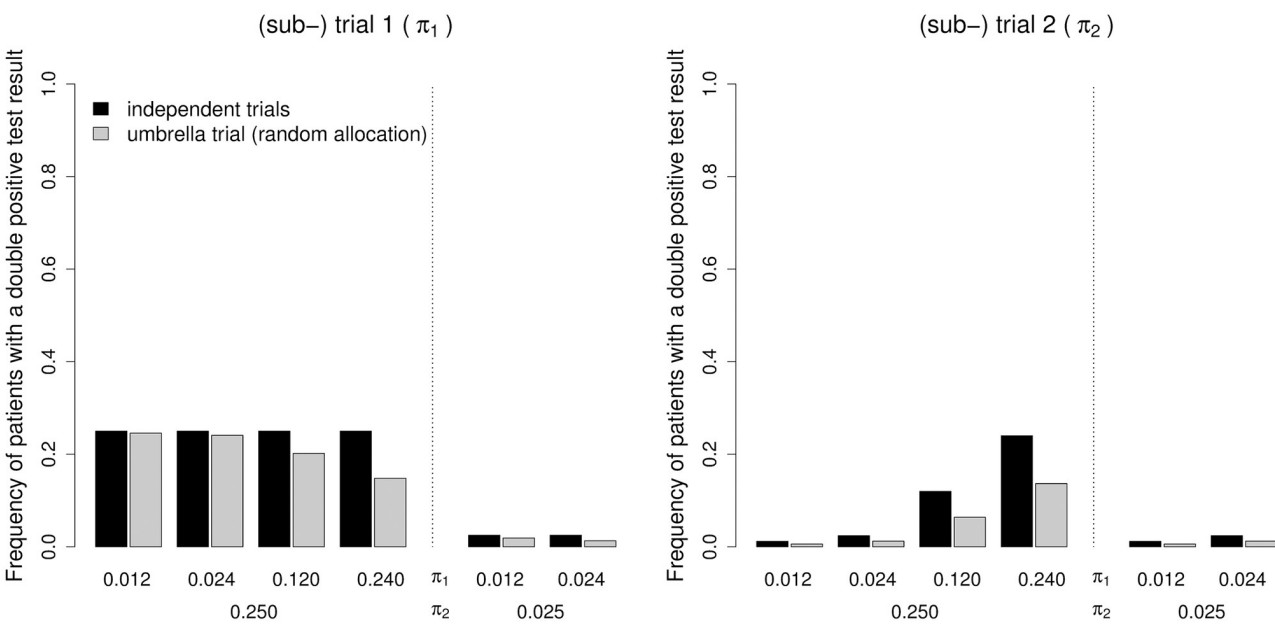

**Fig 3. Proportion of patients with a positive test result for both biomarkers in the independent trial design and in the umbrella design with the random allocation scheme.** $\pi_i$ ($i = 1, 2$) denotes the prevalence of a positive test result for biomarker $B_i$. The underlying formulae are given in Eqs (10) and (12). For each (sub-) trial, the associated prevalence is indicated in brackets next to the (sub-) trial number above the main plot region.

The third aspect are the differences in the expected treatment effect (see Fig 5). Given that they only depend on probabilities and expectations (see Eq (14)) there is no sample size dependency except that the expected effect size $\delta$ has an impact on sample size planning (i.e. larger sample size for a smaller expected effect). If the prevalence of a positive test result is low for biomarker $B_1$, the relative difference is below 2%. If the prevalence of a positive test result is high for both biomarkers, the difference between the expected treatment effect increases to 9 to 21% between trial 1 and subtrial 1 as well as to 3 to 7% between trial 2 and subtrial 2.

## Impact of the pragmatic subtrial allocation scheme and the analysis method

In the following, we use a simulation study and real data from a randomised controlled trial (RCT) to (a) compare the random and the pragmatic subtrial allocation schemes in the umbrella designs and to (b) investigate weighted linear regression as a possible subtrial analysis method that is able to handle the design impact. Details on the weighted linear regression are provided in S2 Note.

### Data sets and analysis setup

**Simulation study.** We revisit the example from the Subsection "Illustration of the analytical derivations" in parts. For the (sub-) trials, we again assume equal sample sizes ($N_1 = N_2 =:$ N) and equal treatment effects ($\delta_1 = \delta_2 =: \delta$). The biomarker status- and (sub-) trial-specific outcomes are summarised in Table 1. The biomarker status distributions in the independent trial design and in the umbrella designs are estimated simulation-based as the mean proportions across 10, 000 simulation runs and then taken as fixed expectation across the simulation

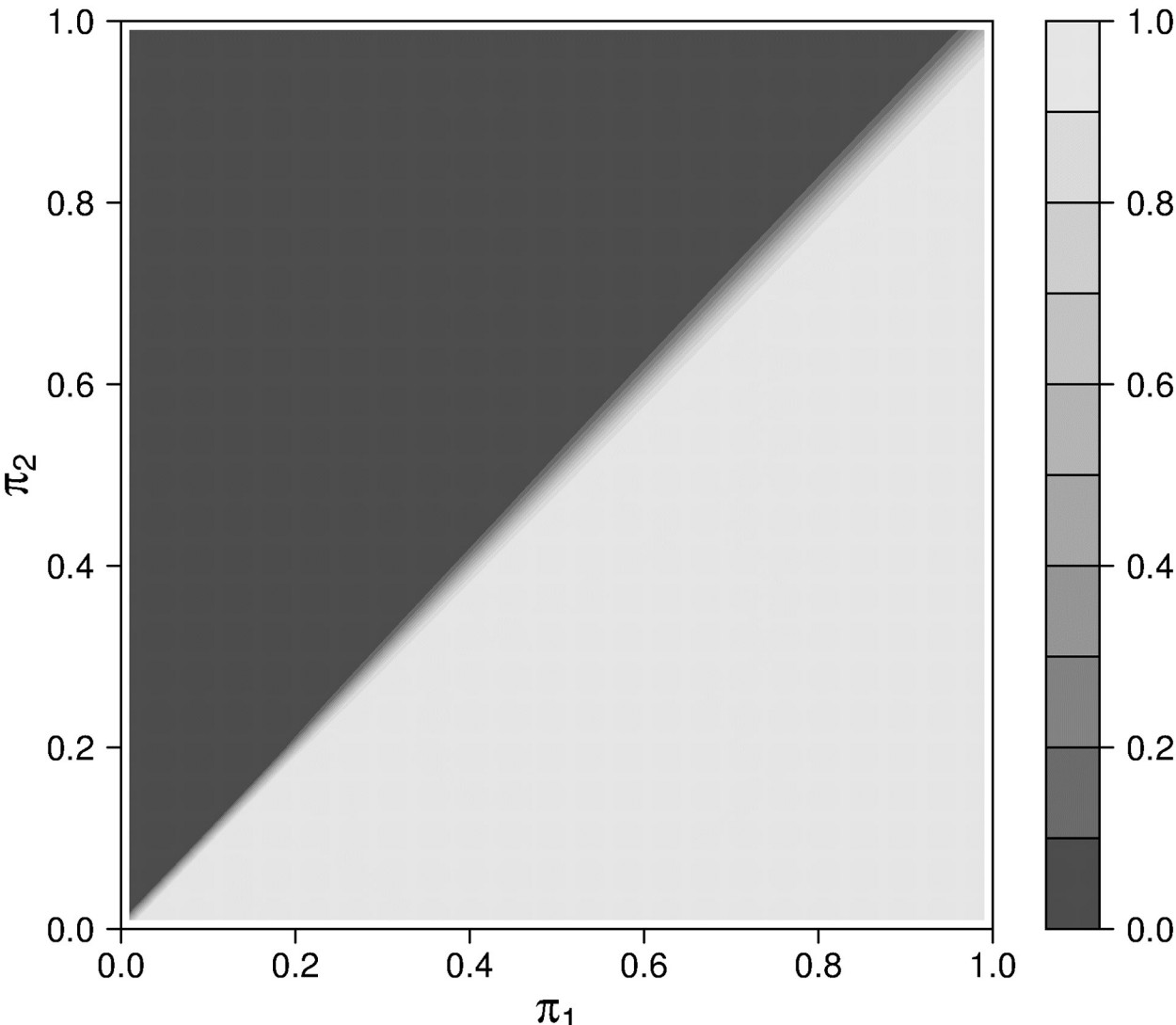

**Fig 4. Probability that subtrial 1 closes earlier than subtrial 2 of the umbrella design with the random allocation scheme for varying prevalence of the two biomarkers.** $\pi_i$ ($i$ = 1, 2) denotes the prevalence of a positive test result for biomarker $B_i$. The underlying formula is given in Eq (8).

runs. The regression weights are calculated based on these a priori fixed distributions according to Equation (S25).

To compare the subtrial allocation schemes, we only consider biomarkers with a high prevalence of a positive test result, i.e. the prevalence tuples (0.12, 0.25) and (0.24, 0.25). Again, we assume for the expected treatment effect in the independent trials

$$\delta = \begin{cases} 0.57 & \text{for } N = 100 \\ 0.36 & \text{for } N = 250 \end{cases}$$

We compare the number of patients needed to be screened and the biomarker status distribution. For the number of patients needed to be screened, we again consider the efficiency

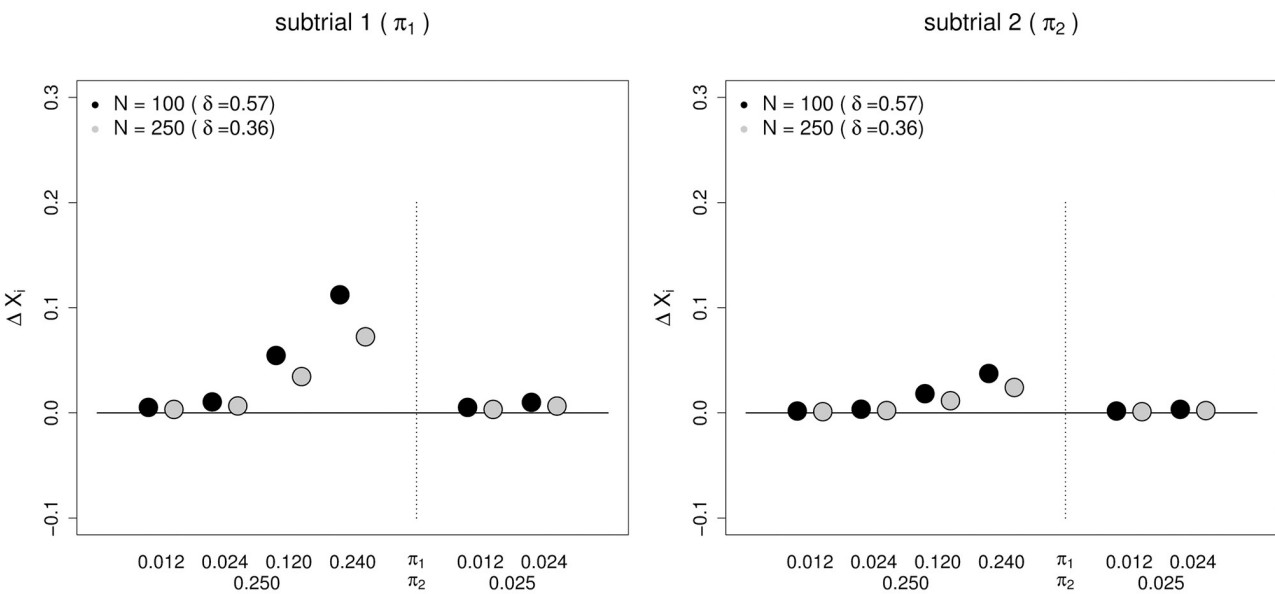

**Fig 5. Difference in the estimated treatment effects between an independent trial and the corresponding umbrella subtrial with the random allocation scheme.** The impact of the biomarker status is larger in (sub-) trial 1. The difference in the treatment effect estimate ($\Delta X_i$, $i = 1, 2$) is given in Eq (14). The treatment effect estimate of subtrial $i$ equals $\delta + \Delta X_i$. N denotes the (sub-) trial size and $\delta$ the true treatment effect in the corresponding independent trial. $\pi_i$ denotes the prevalence of a positive test result for biomarker $B_i$. For each subtrial, the associated prevalence is indicated in brackets next to the (sub-) trial number above the main plot region.

measure of a trial design describing the ratio of discarded (screened but not included) to included patients.

To compare both (sub-) trial analysis methods (un-weighted and weighted mean group differences), we focus only on the (sub-) trial size N of 100, the expected treatment effect $\delta$ of 0.57 in the independent trials and the prevalence tuple (0.12, 0.24). The difference in effect size estimates is the difference between the mean estimated group difference and $\delta$ across all simulation runs for each (sub-) trial.

The simulation study comprises three main steps: (1) screening and (sub-) trial allocation, (2) treatment arm allocation and individual outcome assessment and (3) (sub-) trial analysis. In step (1), we simulate the individual biomarker status $(B_1^+, B_2^+)$ according to the expected biomarker status distribution. We then allocate the patient to the appropriate (sub-) trial or discard the patient if necessary. Once all (sub-) trials are closed—i.e. step (2) –, we randomly assign (with equal probability) the patients to the (sub-) trial-specific experimental or standard treatment arm. Then, we simulate the individual patient's outcome as normally distributed with a standard deviation of 1 and a mean according to biomarker status and (sub-) trial (Table 1). Finally in step (3), we analyse the (sub-) trials separately. We ran the simulation 1, 000 times and report means as summaries of the runs. All simulations and analyses were done with R (version 3.4.1). We applied the function `lm()` from the `stats` package for the weighted linear regression [12].

**Real data application.** We use data of an RCT to mimic the independent as well as the umbrella trial designs given in the Subsections "Independent trial design" and "Umbrella trial design". The underlying RCT is MAXSEP [13]; we refer to the original publications for details. In short, MAXSEP was a randomized, open-label, multi-centre, parallel group trial in which patients with severe sepsis or septic shock were randomised to receive either moxifloxacin and meropenem or only meropenem. The primary endpoint was the mean sequential organ failure

**Table 2. Characteristics of the patients in the reduced data set in the real data application—Overall as well as treatment group-specific.**

| Characteristic | Overall | Treatment 1 | Treatment 2 |
|---|---|---|---|
| Number of patients | 359 | 179 | 180 |
| Baseline | | | |
| Mean SOFA score | 7.9 (3.9) | 8.1 (4.0) | 7.7 (3.8) |
| Lactate, in mmol/l | 2.7 (1.6, 4.7) | 2.6 (1.6, 4.4) | 2.7 (1.6, 4.8) |
| CRP, in mg/l | 199.0 (111.8, 288.1) | 200.0 (110.7, 303.0) | 198.5 (112.9, 271.1) |
| Biomarkers | | | |
| Lactate > 2 mmol/l | 229 (63.8%) | 113 (63.1%) | 116 (64.4%) |
| CRP > 128 mg/l | 247 (68.8%) | 123 (68.7%) | 124 (68.9%) |
| $\phi$: lactate—CRP | -0.09 | -0.14 | -0.05 |

Treatment 1 corresponds to the treatment with moxifloxacin and meropenem and treatment 2 to the treatment with only meropenem. The biomarker dependence is quantified by the $\phi$ coefficient. For the mean SOFA score, mean and standard deviation are reported. For the remaining characteristics, median accompanied by the first and third quartile or absolute and relative frequencies are provided. Abbreviations: CRP—C-reactive protein, SOFA—sequential organ failure assessment.

assessment (SOFA) score [14] over (maximal) 14 days after randomisation. The SOFA score ranges between 0 and 24 points and larger values indicate worse outcome. 551 patients were included in the final analysis.

As in the original data set, we compare the mean SOFA score between the two treatment groups. For illustration of the considered trial designs, we use two biomarkers frequently investigated in sepsis research: lactate and C-reactive protein (CRP). The biomarker-specific positive test results are defined as

baseline lactate value >2 mmol/l for biomarker 1

baseline CRP value >128 mg/l for biomarker 2

according to [15, 16]. Note that these cut-off values were derived for other research questions than therapeutic studies. We exclude patients with missing values in one of the above defined variables from analysis. An overview of the reduced data is given in Table 2. The estimated mean difference (with 95% confidence interval) in the mean SOFA score between the treatment groups in the reduced data set is 0.35 (−0.45, 1.16) which is similar to the original publication [13].

From this reduced data set, we sample 1, 000 (bootstrap) samples with replacement (i.e. we simulate data sets from the empirical distribution of the trial data). For the (sub-) trials, we again assume equal sample sizes ($N_1 = N_2 =: N$) with $N \in \{50, 100, 250\}$, treatment groups of the same size and apply the design definitions from the Subsections "Independent trial design" and "Umbrella trial design". Both umbrella trials (with its subtrials), i.e. one with the random and one with the pragmatic allocation scheme, and the corresponding independent trials are built from the same bootstrap sample so that these designs are directly comparable. The (sub-) trials are analysed with un-weighted and weighted linear regression. The regression weights are 1 in the independent trial design. The regression weights in case of the umbrella designs are calculated based on the proportions observed in the current umbrella trial (see S2 Note for details). For design comparison, we again report (i) the ratio of discarded (screened but not included) to included patients, (ii) the proportions of patients with a positive test result for both biomarkers and (iii) the regression coefficients (as estimated treatment effect). Furthermore, we compare the means across the bootstrap samples with the corresponding analytical

**Table 3. Ratio of discarded to included patients in the independent trial design as well as in both umbrella designs.**

| $(\pi_1, \pi_2)$ | Independent design | Umbrella design | |
| --- | --- | --- | --- |
| | | random allocation | pragmatic allocation |
| (0.12, 0.25) | 5.2 | 3.4 | 3.2 |
| (0.24, 0.25) | 3.1 | 1.4 | 1.3 |

$\pi_i$ ($i$ = 1, 2) denotes the prevalence of a positive test result for biomarker $B_i$. A "discarded patient" is a patient that was screened but not included in a (sub-) trial. The ratio is derived as $N_{screen} - 2N$ divided by 2N. $N_{screen}$ denotes the mean number of screened patients across the simulation runs. There are 2N included patients in each design. The ratios for the independent trial design and for the umbrella design with the random allocation scheme correspond to the ratios given in the Subsection "Illustration of the analytical derivations" (Fig 3).

expectations from the Section "Impact of the umbrella design with the random allocation scheme" applied to the reduced data set.

## Results

**Simulation study.** In an umbrella trial, the application of the pragmatic allocation scheme is more efficient than the application of the random allocation scheme (Table 3). The relative decrease of the ratio of the number of discarded patients to the number of included patients is about 5 to 8%. However, the choice of the subtrial allocation scheme within the umbrella design has a much smaller impact on the number of screened patients than the switch from the independent trial design to an umbrella design. In the latter case, the reductions may range between 33 and 50% depending on prevalence constellations.

In an umbrella trial, the application of the pragmatic allocation scheme may lead to even larger differences in the biomarker status distribution compared to the independent trials (Table 4). Comparing the umbrella designs, the distributions are similar for the prevalence tuple (0.24, 0.25). In contrast to the random allocation scheme, the application of the pragmatic allocation scheme for the tuple (0.12, 0.25) leads to an inclusion of all patients with a double positive test result in subtrial 1 at the cost of missing these patients completely in subtrial 2. Consequently, the distribution of subtrial 1 is identical to the distribution of trial 1.

In the independent trial design, both the un-weighted and weighted estimated group differences reflect the true treatment effect (Fig 6). In the umbrella design with the random allocation scheme, weighted linear regression results in estimates closer to the true treatment effect of the corresponding independent trial. However, when relying on estimated un-weighted

**Table 4. Proportion of patients with a positive test result for both biomarkers in the independent trial design as well as in both umbrella designs.**

| $(\pi_1, \pi_2)$ | (Sub-) trial | Independent design | Umbrella design | |
| --- | --- | --- | --- | --- |
| | | | random allocation | pragmatic allocation |
| (0.12, 0.25) | 1 | 0.25 | 0.20 | 0.25 |
| | 2 | 0.12 | 0.06 | 0.00* |
| (0.24, 0.25) | 1 | 0.25 | 0.15 | 0.16 |
| | 2 | 0.24 | 0.14 | 0.12 |

$\pi_i$ ($i$ = 1, 2) denotes the prevalence of a positive test result for biomarker $B_i$. For the independent trial design and the umbrella design with the random allocation scheme, the distributions correspond to those provided in the Subsection "Illustration of the analytical derivations" (Fig 3). The proportions are based on mean proportions across the simulation runs.

* in case of rounding to four digits: 0.0002

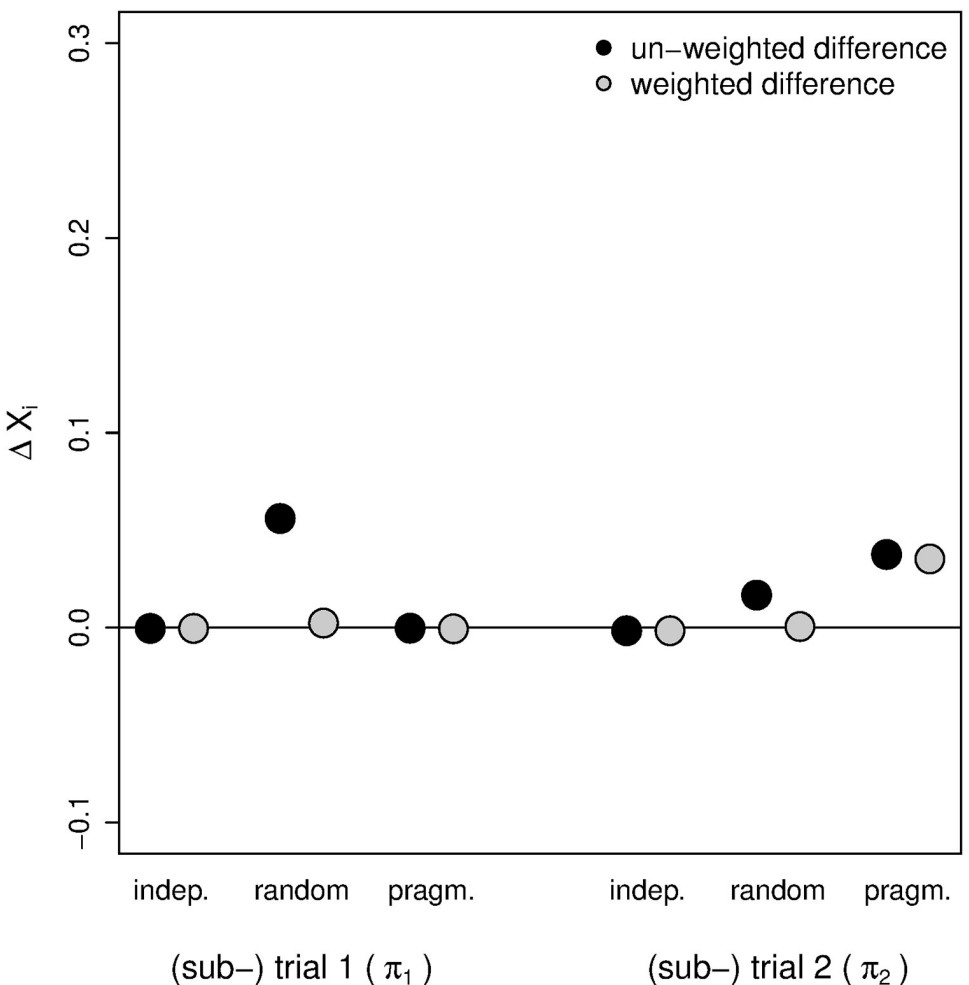

**Fig 6. Difference in the estimated treatment effects between an independent trial and the corresponding umbrella subtrial in the application of different analysis methods in the simulation study.** The weights for the weighted linear regression are given in equation (S25) in S2 Note. The biomarker status impact on the treatment response is larger in (sub-) trial 1. $\Delta X_i$ ($i = 1, 2$) is the difference between the mean treatment effect estimate across the simulation runs and $\delta$. $\delta$ denotes the true treatment effect in the corresponding independent trial. $\pi_i$ denotes the prevalence of a positive test result for biomarker $B_i$. For each subtrial, the associated prevalence is indicated in brackets next to the (sub-) trial number below the main plot region. indep.: independent trial design, random: umbrella trial design with the random allocation scheme, pragm.: umbrella trial design with the pragmatic allocation scheme.

mean group differences, the difference between the designs is larger in subtrial 1 in which the effect of the $B_2$ value on the patient's response to the experimental treatment is stronger than the respective effect in subtrial 2. Applying the pragmatic allocation scheme, subtrial 1 and trial 1 results are identical because all patients with a double positive test result are assigned to subtrial 1. Consequently, the weighting has no effect in subtrial 2 given that there are no patients with a double positive test result in this subtrial.

**Real data application.**   Results of five exemplary (bootstrap) samples are provided in S1 Table. Overall and as already demonstrated above, the umbrella design can reduce the proportion of discarded (screened but not included) patients (S1 Fig) as well as shift the biomarker status distribution (S2 Fig) compared to the corresponding independent trial design. This design impact depends on the chosen subtrial allocation scheme. The impact of the shift in the

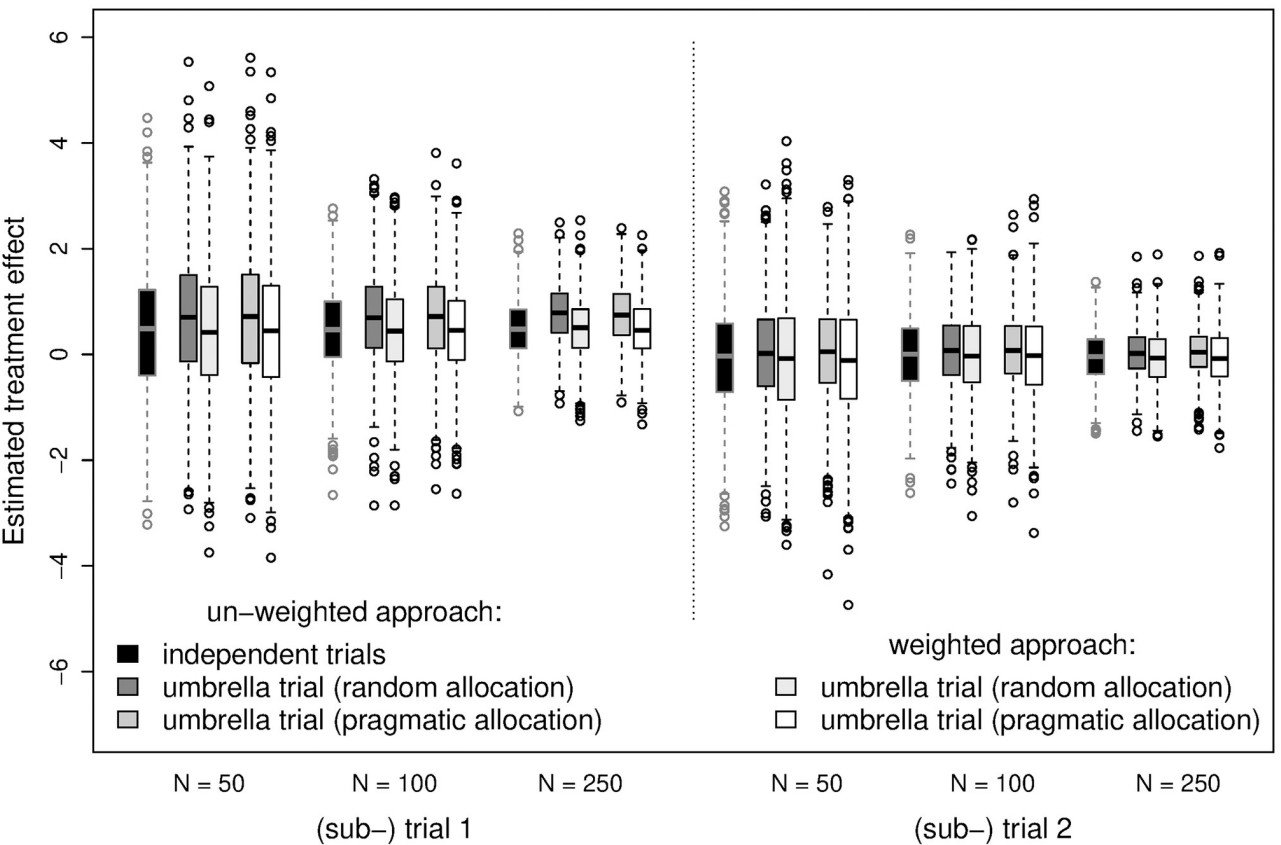

**Fig 7. Estimated treatment effects in the independent trial and in both umbrella trial designs in the application of different analysis methods in the real data application.** The estimated treatment effect corresponds to the regression coefficient. The calculation of the weights for the weighted linear regression are given in S2 Note. The trial size *N*, the trial design, the respective subtrial and the analysis method are indicated. The distribution across the 1, 000 bootstrap runs are provided.

biomarker status distribution on the estimated treatment effect is given in Fig 7. Overall, the weighting reduces the difference between the estimate from an umbrella subtrial and that from the corresponding independent trial compared to the un-weighted approach. Note that, in a particular trial, the weighting can increase the difference between the results from an umbrella subtrial and those from the corresponding independent trial and the un-weighted result from the umbrella trial can be close to the result from the independent trial design (see also S1 Table). Furthermore, the weighting (i.e. a deviation from equal weights) increases the variability of the estimate and, hence, reduces the power. This is more pronounced in subtrial 2 in this real data application. This aspect is directly related to the aspect of the error probabilities (Subsection "Estimated treatment effect"). It should further be noted that due to the weights being estimated from the screening process and plugged-in in the weighted linear regression model, type I error may be affected for smaller sample sizes (in contrast to a priori fixed weights).

A second aspect in this real data application was the comparison of the analytical deductions and the bootstrap means for the umbrella design with the random allocation scheme and the corresponding independent trial design. If the dependence (correlation) between the biomarkers (here: $\phi = -0.09$ from the reduced data set) is known, the expected number of discarded (screened but not included) patients (Subsection "Number of patients needed to be screened") and the expected number of patients with a positive test result for both biomarkers

(Subsection "Biomarker status distribution") are similar to the mean of the bootstrap runs (S2 (a) and S2(b) Fig). If independence of the biomarker is assumed, the means from the bootstrap runs are overestimated. In case of the expected estimated difference in the treatment effects, the assumed true treatment effect was based on the effect observed in the reduced data set (Subsection "Estimated treatment effect"). This might have introduced the differences between expectation and bootstrap mean (S2c Table).

## Discussion

In order to quantify the properties of the umbrella design compared to the independent trial design, we first investigated the impact of an umbrella design with two subtrials on several aspects of trial conduct. Secondly, we analysed the influence of the umbrella subtrial allocation scheme and finally proposed a possible solution to handle the consequences of the application of an umbrella design. Regarding our first objective, we identified a clear efficiency benefit—in terms of patients needed to be screened—of umbrella trials when applying the random subtrial allocation scheme. We showed that the biomarker status distribution could differ between an independent trial and the corresponding umbrella trial designs. This is caused by the screening platform of umbrella trials where subtrials compete for patients with a positive test result for multiple biomarkers. Such patients are eligible for multiple subtrials but can often only be allocated to one subtrial. Consequently, such patients may be underrepresented in umbrella subtrials compared to the corresponding design with independent trials. If we assume that the targeted treatments operate in a particular biological pathway, having a positive test result for multiple biomarkers would imply the necessity to apply multiple treatments; if this is not the case outcomes of patients might differ between those patients with multiple positive test results and those with a positive test result for only one biomarker (see also e.g. [17]). The consequence of such a biomarker status-dependent treatment effect is an observable difference in (overall) treatment effect estimates between independent trials and the respective umbrella subtrials. In general, the estimated treatment effect in an umbrella subtrial can be larger or smaller than in the respective independent trial. As a result, statistical (error) probabilities—i.e. the comparison-wise statistical power and type I error—can also be affected.

Regarding our second objective, the comparison of two allocation schemes (random versus pragmatic) for umbrella trials, we observed that the pragmatic allocation scheme further reduces the number of patients needed to be screened. This benefit may come at the expense that no patients with a positive test result for multiple biomarkers are included in one of the subtrials. Consequently, the difference between estimated treatment effects of independent trials and the respective umbrella subtrials may even be larger than in the setting of the umbrella design with random allocation scheme. However, the impact of the subtrial allocation scheme is of minor importance compared to the impact of the screening platform itself.

Finally, we demonstrated that weighted linear regression in combination with the random allocation scheme for the umbrella subtrials might be applied to get treatment effect estimates that are similar for umbrella subtrial and the corresponding independent trial.

In this investigation we quantified the impact of umbrella designs on results of clinical trials. The reduction of patients needed to be screened and the changes in the biomarker status distribution is induced by the design definition (e.g. [1, 2]). The efficiency has also been previously evaluated for several umbrella (platform) designs [18]. It has been stated earlier that results from umbrella trials must be interpreted with care if the outcome of patients tested positive for multiple biomarkers (possibly over- or underrepresented in subtrials) differs from those with a positive test result for fewer (or a single) biomarker. There, it was mentioned that their influence might be minimal if the number of such patients is small but might be

substantial if it would be vice versa [19]. These statements are in accordance with our results. Hence, one must account for this impact and acknowledge it already in the statistical analysis plan [20].

## Translation of our results to other trial design variants

These design differences will also be present with a weighted random, a hierarchical or a Bayesian allocation scheme that are already applied in umbrella trials [7, 21–24]. Thus, our principle findings also apply to these umbrella trial variants. At a more general level, the umbrella design is an example for a master protocol design. Another master protocol design is the basket design, e.g. [1, 2]. In a basket design, one biomarker-driven experimental treatment is investigated for several diseases. Thus, each disease defines a subtrial. Consequently, our observations for the umbrella design should also translate to basket trials by switching the role of biomarker and disease when defining the subtrials—although patients exhibiting more than one disease are probably rarer than patients with positive test results for more than one bio-marker and might not be included in a basket trial at all. In general, the findings can be gener-alised to other platform designs [25] and master protocol designs whenever their subtrials compete for eligible patients. An overview of possible analysis approaches for these designs is provided for example in [26].

## Analysis strategies and modelling approaches in umbrella designs

In order to evaluate the impact of the joint screening platform itself keeping the analysis strat-egy of independent trials, we followed the approach of analysing each subtrial separately, although comparisons between the subtrials or to an overall/shared control arm including patients exhibiting negative test results for all biomarkers under investigation receiving the standard treatment (see Fig 1) are other possible analysis approaches to analyse umbrella designs. Depending on the chosen (subgroup) comparison between subtrials, the comparison might not be affected by the shift in the biomarker status distribution introduced by the umbrella design (e.g. within patients with a specific biomarker status). The shared control arm is a tool to further reduce the number of discarded (screened but not included) patients. How-ever, it is based on the assumption that the patient's response to the standard therapy is inde-pendent of its biomarker status. A shared control arm can introduce questions about data integrity [27–29].

One modelling strategy to address the shift in treatment effect estimates that could arise in umbrella trials is to apply weighted linear regression. Obviously, weighting can only be applied if there are patients to weight and may be questionable in the case of building the weighting on very few patients with a specific biomarker status. Thus, weighting will require a minimum sample size for the strata. These aspects especially apply to the pragmatic allocation scheme where small proportions of patients with a specific biomarker status can be expected more fre-quently than with the random allocation scheme. Consequently, the impact of each observa-tion on the estimates can differ between the designs and these differences might even be aggravated by the weighting. Furthermore, it should be noted that it is well known that the weighting induces a loss in efficiency if not all weights are 1 [30] which is also supported by the results of the real data application. Whether this loss in efficiency is of importance must be assessed individually for each trial and weighed against the possible gain in consistency [31–33].

It should be noted that the properties of weighted regression (including a correct type I error) is based on weights that would be fixed in advance. If the biomarker distribution in the overall patient population is considered to be known in advance, weighted regression with

pre-specified weights would be a preferable option to be investigated further. The proposed plug-in procedure using the estimated proportions of the combined biomarker status from the screening process would rather rely on asymptotic properties but allows for an approximate solution in a setting with unknown prevalence of biomarker status combinations. Obviously, as indicated above, a very low number of double positive patients would hamper the plugged-in approach.

Other modelling approaches comprise for example regression models with main and interaction terms or an analysis stratified by the biomarker status. In general, one should carefully distinguish between the planned, confirmatory analyses and post-hoc sensitivity analyses. If a biological interaction between a co-expressed biomarker and the treatment is expected, this interaction should be modelled and even better already be addressed at the trial planning stage. The same applies to possible interactions between the biomarkers. Known interactions should be addressed by design; data-driven explorations are necessary but would be declared as sensitivity analyses. Further approaches for umbrella designs are provided for example in [26].

## Strength and limitations of this study

Obviously, this study also has several limitations: (i) In the illustrative example and in the simulation study, we defined the biomarker status-dependent treatment effect under the assumption that a patient with a positive test result for multiple biomarkers would show a smaller treatment effect than a patient with a positive test result for only one biomarker. This assumption is based on the idea that patients with a positive test result for multiple biomarkers would require multiple treatments (e.g. [17]). If these patients only receive single treatments, their outcomes might be smaller. However, our observations also apply in case the targeted treatment has a different treatment effect, e.g. vice versa to the effect assumed in this investigation. Then, the estimated treatment effect can be smaller in the umbrella subtrial than in the corresponding independent trial. (ii) The evaluated treatment effects, an umbrella design with just two independent biomarkers and no option to stop or add subtrials is likely unrealistic. However, the treatment effects were chosen as one would do in sample size calculations. Dependence between the biomarkers would only shift the biomarker status distribution in the disease population. More biomarkers will change and more subtrials or adding/stopping subtrials will increase the likelihood of positive test results for multiple biomarkers. In general, our arbitrary or unrealistic choices will have no influence on the general message of the paper that the estimated treatment effect depends on the selected design—independent trials or umbrella subtrials—and that this must be handled with care at the planning stage of such a trial (e.g. [34]). (iii) In the simulation study, we decided not to pursue the pragmatic allocation scheme in the umbrella design for the case of biomarkers with a low prevalence of a positive test result. This approach seems justified as the results for biomarkers with a high prevalence can be up-scaled to the low prevalence setting. If the prevalence of positive test results for both biomarkers is similar, the application of the pragmatic allocation scheme and of the random allocation scheme in an umbrella design induce similar subtrial structures. Applying the pragmatic allocation scheme in an umbrella design, the largest efficiency gain in terms of number of patients needed to be screened is achieved if all patients with a positive test result for both biomarkers are allocated to one subtrial but this induces the largest possible shift in the biomarker status distribution in the second subtrial compared to the corresponding independent trial. However, in case a positive test result for at least one biomarker is a low prevalent event, patients with a positive test result for both biomarkers are also low prevalent and the absolute gain in efficiency is less pronounced compared to the setting where positive test results are

high prevalent for both biomarkers. (iv) We considered a normally distributed endpoint while on-going or published umbrella trials focussed on time-to-event, e.g. [23], or binary endpoints, e.g. [22]. Given our general observation of the umbrella design impact on treatment effect estimates, this difference in estimates will also be present for other endpoints. (v) Although we consider that a weighted approach (either with fixed or empirical weights) could be a solution to the difficulties in effect estimation and resulting potential type-1 error inflation in the presence of an interaction between treatment and biomarker status, we did not investigate empirical type I error rates and statistical power of the weighted approach in more detail. (vi) In general, regarding the control of the overall (study-wise) type I error rate, different approaches are currently discussed for umbrella trials, as e.g. the reliance on the biomarker-defined population-wise error rate [35] instead of the study-wise error rate. Another proposal [36] is to investigate the single experimental treatment only if the complete treatment strategy shows an effect with respect to the endpoint. However, this approach does not allow for starting new subtrials once the umbrella trial is running and treatment-by-biomarker interaction may also have an impact on the properties of such a procedure. Either way, the issue of multiplicity in umbrella trials is beyond the scope of this article. (vii) We could not present an application of the methods to a real umbrella trial given that there were no published patient-level data from umbrella designs available to us. To address this limitation to some extent, we relied on data from a randomised controlled trial and used bootstrapping to mimic umbrella subtrials and the corresponding independent trials. Furthermore, we illustrated the analytical derivations by real data parameter choices from oncology and pharmacogenomics and with assumptions for common sample size calculations. Additionally, we showed how to calculate the weights in order to mimic the biomarker status distribution for the corresponding independent trial design.

## Conclusions

In summary, we identified several benefits of the umbrella design such as much fewer patients needed to be screened relative to a setting of running independent trials. Conversely, the price to be paid are treatment effect estimates that may deviate from those obtained from the corresponding independent trials. This difference increases with increasing proportions of patients with a positive test results for multiple biomarkers. As a starting point, we show that the random allocation scheme combined with a weighted regression analysis could address this issue. As a result of the more frequent use of umbrella designs, more research in order to practically inform the analyst is needed.

## Supporting information

**S1 File. R source files.** Code to reproduce the results of the analytical calculations and of the simulation study of this article.
(ZIP)

**S1 Note. Umbrella trial design with the random allocation scheme: Number of patients needed to be screened and biomarker status distribution.**
(PDF)

**S2 Note. Weighted linear regression as (sub-) trial analysis method.**
(PDF)

**S3 Note. Additional results from the real data application.**
(PDF)

**S1 Fig. Ratio of discarded to included patients in both umbrella designs and the corresponding independent trial design.** The trial size $N$ is indicated. A "discarded patient" is a patient that was screened but not included in a (sub-) trial. The ratio is derived as $N_{screen} - 2N$ divided by $2N$. $N_{screen}$ denotes the number of screened patients in a bootstrap run. There are $2N$ included patients in each design. The distribution across the 1, 000 bootstrap runs are provided.
(TIF)

**S2 Fig. Proportion of patients with a positive test result for both biomarkers in both umbrella designs and in the corresponding independent trial design in the real data application.** The trial size $N$ and the respective subtrial are indicated. The distribution across the 1, 000 bootstrap runs are provided.
(TIF)

**S1 Table. Results from five exemplary (bootstrap) samples in the real data application for the independent trial design and both umbrella designs.**
(PDF)

**S2 Table. Application of the analytical calculation formulae to the real data and its comparison with the bootstrap means for the independent trial design and the umbrella design with the random allocation scheme.**
(PDF)

## Acknowledgments

The authors would like to thank the patients of the VISEP and MAXSEP trials for their contribution to clinical sepsis research and the SepNet Study Group to kindly provide the data for the real data application. Within this context, the authors want to express their special thanks to Evelyn Trips and Holger Bogatsch who were part of the team running the original analysis and prepared the data set for the real data application in this article, respectively.

## Disclaimer

Views expressed in this publication are the author's (NB) personal views and not necessarily the views of BfArM (Federal Institute for Drugs and Medical Devices, Bonn, Germany).

## Author Contributions

**Conceptualization:** Miriam Kesselmeier, André Scherag.

**Formal analysis:** Miriam Kesselmeier, Norbert Benda.

**Investigation:** Miriam Kesselmeier, Norbert Benda.

**Methodology:** Miriam Kesselmeier, Norbert Benda, André Scherag.

**Software:** Miriam Kesselmeier.

**Validation:** Miriam Kesselmeier.

**Visualization:** Miriam Kesselmeier.

**Writing – original draft:** Miriam Kesselmeier.

**Writing – review & editing:** Miriam Kesselmeier, Norbert Benda, André Scherag.

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
