## [Decision Letter · Decision Letter 0]

17 Jun 2020

PONE-D-19-35136

Effect size estimates from umbrella designs: handling patients with a positive test result for multiple biomarkers using random or pragmatic subtrial allocation

PLOS ONE

Dear Dr Kesselmeier,

Thank you for submitting your manuscript to PLOS ONE. After careful consideration, we feel that it has merit but does not fully meet PLOS ONE’s publication criteria as it currently stands. Therefore, we invite you to submit a revised version of the manuscript that addresses the points raised during the review process.

Please add a subsection that describes "what this study added to the current knowledge."Further comparisons with relevant studies still need to be included in the discussion section. Please add clear and separate subsections that describe "strength and the limitations of this study" The authors discussed the limitations of the study (which still needs to be placed in a separate subsection); however, the future work plan was not mentioned and how the authors plan to overcome the limitations of this study in the future work.Please submit your revised manuscript by 2^nd^ August, 2020. If you will need more time than this to complete your revisions, please reply to this message or contact the journal office at plosone@plos.org. Please include the following items when submitting your revised manuscript:A rebuttal letter that responds to each point raised by the academic editor and reviewer(s). You should upload this letter as a separate file labeled 'Response to Reviewers'.A marked-up copy of your manuscript that highlights changes made to the original version. You should upload this as a separate file labeled 'Revised Manuscript with Track Changes'.An unmarked version of your revised paper without tracked changes. You should upload this as a separate file labeled 'Manuscript'.

We look forward to receiving your revised manuscript.

Kind regards,

Professor Khaled Khatab, Ph.D.

Academic Editor

PLOS ONE

Journal Requirements:

Reviewers' comments:

Reviewer's Responses to Questions

**Comments to the Author**

1. Is the manuscript technically sound, and do the data support the conclusions?

Reviewer #1: Yes

2. Has the statistical analysis been performed appropriately and rigorously? 

Reviewer #1: Yes

3. Have the authors made all data underlying the findings in their manuscript fully available?

Reviewer #1: Yes

4. Is the manuscript presented in an intelligible fashion and written in standard English?

Reviewer #1: Yes

5. Review Comments to the Author

Reviewer #1: Following from the previous three reviewers of this manuscript, I can confirm that the authors have significantly revised the manuscript in accordance with the concerns raised by the previous reviewers.

The authors have demonstrated their originality, ingenuity and expertise in the presentation of the main idea of the papers. The statistical methods applied are appropriately applied and are relevant to the objectives of the study with detailed supporting information included.

However, the areas I feel the authors should review are minor. For example;

In Line 46, I suggest the following sentence: "The aim of this study is three-fold" be reworded as: "The objectives of this study are in three-fold (or are fundamentally three)".....

In line 70, the parameter (or coefficient), ϕ (phi) should rather be referenced "correlation measure" and not dependency measure as used by the authors.

Though figures A1 and A2 like other figures were referenced in the manuscript, I expected the authors to give brief discussion on the figures, in terms of the information they want the readers to derive and how do such details support to their results.

Finally, I suggest that the authors check to see they explain what each parameters used in the equations uniquely represent to enhance the comprehension and flow of their lay audience.

6. PLOS authors have the option to publish the peer review history of their article (what does this mean?). If published, this will include your full peer review and any attached files.

**Do you want your identity to be public for this peer review?** For information about this choice, including consent withdrawal, please see our Privacy Policy.

Reviewer #1: No

---

## [Author Response · Author response to Decision Letter 0]

13 Jul 2020

Point-by-point response to the editor’s and reviewer’s comments

Academic editor

Please add a subsection that describes "what this study added to the current knowledge."

RESPONSE: 

Thank you for this comment. We included this paragraph in the Discussion as fourth paragraph after our summary of our investigation (first three paragraphs). 

Further comparisons with relevant studies still need to be included in the discussion section. 

RESPONSE: 

We repeated our literature search to identify the current evidence and extended the discussion where necessary. The newly introduced literature is highlighted in red (in the text body, not in the references) in the manuscript version with tracked changes.

Please add clear and separate subsections that describe "strength and the limitations of this study"

RESPONSE: 

We included the subsection heading “Strength and limitations of this study” and additional headings for an easier navigation within the discussion. 

The authors discussed the limitations of the study (which still needs to be placed in a separate subsection); however, the future work plan was not mentioned and how the authors plan to overcome the limitations of this study in the future work. 

RESPONSE: 

We agree that the future work is not directly described. We contrasted the limitations with the strengths and discussed the impact of the limitations on the conclusions from our investigation. 

Our research was just the first step to quantify the possible impact of an umbrella design on clinical trial results. This point need further investigation, which we now state in the revised version of the manuscript as suggested by the reviewer. As we did not want to leave the reader without any idea of a possible solution, we applied weighted linear regression as an obvious solution. However, it is unclear to which extent this approach can be used in practice. Consequently, there is more work to be done for a solid advice. 

We originally did not include this statement, as we wanted to avoid those typical last sentences stating that further work is needed. Nevertheless, we added such a sentence in the revised version of the manuscript, but would leave it to the editor whether to include it or not. 

Reviewer #1

Following from the previous three reviewers of this manuscript, I can confirm that the authors have significantly revised the manuscript in accordance with the concerns raised by the previous reviewers.

The authors have demonstrated their originality, ingenuity and expertise in the presentation of the main idea of the papers. The statistical methods applied are appropriately applied and are relevant to the objectives of the study with detailed supporting information included. 

RESPONSE: 

We thank the reviewer for his thorough review and the helpful suggestions. We provide a point-by-point response below. Please note that we had to adapt the referencing to the sections, as there is no section numbering allowed.

However, the areas I feel the authors should review are minor. For example;

In Line 46, I suggest the following sentence: "The aim of this study is three-fold" be reworded as: "The objectives of this study are in three-fold (or are fundamentally three)"..... 

RESPONSE: 

Done. Thank you. We rephrased the sentence. 

In line 70, the parameter (or coefficient), ϕ (phi) should rather be referenced "correlation measure" and not dependency measure as used by the authors. 

RESPONSE: 

We slightly modified the wording and write throughout the manuscript “correlation (dependency) measure” to emphasise that we use the correlation for the definition of the dependency between the biomarkers. Otherwise, it might be confusing as we talk about (in)dependent biomarkers throughout the manuscript.

Though figures A1 and A2 like other figures were referenced in the manuscript, I expected the authors to give brief discussion on the figures, in terms of the information they want the readers to derive and how do such details support to their results. 

RESPONSE: 

The results presented in Fig S1 and S2 are, as stated in the manuscript, in line with the results from the analytical deductions as well as the simulation study. If we extend the presentation in the results, the description would be similar to those in the previous sections. As we want to avoid reporting similar results several times for the ease of reading, we originally decided to show the results for completeness but comment only very briefly on it. The real data application supports the previous results. The content of the supplemental figures is for the readers a proof of principle for the real data application. 

In the revised version of the manuscript, we tried to write it more clearly by inclusion of the word “already”.

Finally, I suggest that the authors check to see they explain what each parameters used in the equations uniquely represent to enhance the comprehension and flow of their lay audience. 

RESPONSE: 

Thank you for this comment. We rephrased parts of the deductions in the S2 Note to (hopefully) clarify and facilitate reading. Furthermore, we added missing + for the test result indicators in the definition of phi.

---

## [Editor Report · Decision Letter 1]

28 Jul 2020

Effect size estimates from umbrella designs: handling patients with a positive test result for multiple biomarkers using random or pragmatic subtrial allocation

PONE-D-19-35136R1

Dear Dr Kesselmeier,

We’re pleased to inform you that your manuscript has been judged scientifically suitable for publication and will be formally accepted for publication once it meets all outstanding technical requirements.

Within one week, you’ll receive an e-mail detailing the required amendments. When these have been addressed, you’ll receive a formal acceptance letter, and your manuscript will be scheduled for publication.

An invoice for payment will follow shortly after the formal acceptance. To ensure an efficient process, please log into Editorial Manager at http://www.editorialmanager.com/pone/, click the 'Update My Information' link at the top of the page, and double-check that your user information is up-to-date. If you have any billing-related questions, please contact our Author Billing department directly at authorbilling@plos.org.

Kind regards,

Professor Khaled Khatab, PhD.

Academic Editor

PLOS ONE
---

## [Editor Report · Acceptance letter]

3 Aug 2020

PONE-D-19-35136R1 

Effect size estimates from umbrella designs: handling patients with a positive test result for multiple biomarkers using random or pragmatic subtrial allocation 

Dear Dr. Kesselmeier:

I'm pleased to inform you that your manuscript has been deemed suitable for publication in PLOS ONE. Congratulations! Your manuscript is now with our production department. 

Kind regards, 

on behalf of

Professor Khaled Khatab 

Academic Editor

PLOS ONE